# Satellite-Based Wireless Sensor Development and Deployment Studies for Surface Wave Testing

**DOI:** 10.3390/s19204364

**Published:** 2019-10-09

**Authors:** Pengju Xu, Wentao Wang

**Affiliations:** 1Key Lab of Structures Dynamic Behavior and Control of the Ministry of Education, Harbin Institutive of Technology, Harbin 150090, China; xupengju@hit.edu.cn; 2Department of Civil and Environmental Engineering, University of Michigan, Ann Arbor, MI 48109-2125, USA

**Keywords:** wireless sensing, embedded data processing, GPS, surface wave test, Rayleigh wave, compressive sampling, nondestructive testing/evaluation

## Abstract

Although cable-based seismic sensing systems have provided reliable data in the past several decades, they become a bottleneck for large-area monitoring and critical environmental (volcanic eruptions) sensing because of their cost, difficulty in deploying and expanding, and lack of accurate three-dimensional geographic information. In this paper, a new wireless sensing system is designed consisting of a portable satellite device, a self-sustaining power source, a low-cost computational core, and a high-precision sensor. The emphasis of this paper is to implement in low-cost hardware without requirements of highly specialized and expensive data acquisition instruments. Meanwhile, a computational-core-embedded algorithm based on compressive sensing (CS) is also developed to compress data size for transmission and encrypt the measured data preventing information loss. Seismic data captured by the accelerometer sensor are coded into compressive data packages and then transferred via satellite communication to a cloud-based server for storage. Acceleration and GPS information is decrypted by the ℓ_1_-norm minimization optimization algorithm for further processing. In this research, the feasibility of the proposed sensing system for the acquisition of seismic testing is investigated in an outdoor field surface wave testing. Results indicate the proposed low-cost wireless sensing system has the capability of collecting ground motions, transferring data, and sharing GPS information via satellite communication for large area monitoring. In addition, it has a great potential of recovering measurements even with significant data package loss.

## 1. Introduction

Soil layers and rocks are complex, multi-phase, particulate, and discontinuous materials which cannot be described as a simple elastic model with fixed mechanical behavior [1,2,3,4]. The soil layer’s nonlinear and irreversible behavior has attracted the interest of researchers for decades [5,6]. An underground structure, designed and built in the deep soil layer under the surface of earth, is a promising alternative to aboveground buildings for its resistance to severe weather, quiet living space, and energy efficiency due to the natural insulating properties of the surrounding earth [7,8]. Likewise, underground transportation helps a major city become a better place by reducing traffic congestion as well as the overall level of pollution [9]. Although underground structures and transportation relieve the congestion resulting from the rapid increase in population, drawbacks remain. Underground construction activities have been challenged by the complicated mechanical behavior of the soil layers including their various thicknesses and main component [10,11].

There is urgent need of accurate measurements on properties of the soil layer for geological exploration, subway, tunnel, and infrastructures constructions such as high buildings or long bridges [12]. To investigate the mechanical properties of soil, a seismic survey is represented, attempting to image or map the subsurface of the earth by sending sound energy down into the ground and recording the "echoes” that return from the rock layers below [1]. Several tests have been designed to measure or estimate seismic wave velocities in situ with the primary purpose of evaluating the mechanical parameters from the wave velocity, such as cross-hole test, down-hole test, multichannel analysis of surface wave [13], surface wave testing, etc. Surface wave as a non-destructive evaluation method is widely used in geophysics to infer the Rayleigh wave velocity of the subsoil for evaluation of the mechanical characteristics of soil layers.

One common device for seismic survey is the geophone, which converts ground movement (velocity or accelerometer) into a time-history voltage signal to be recorded for the analysis of the structure of the measured earth. The geophone has been playing an important role in seismology measurement. While current cable-based seismic acquisition systems including geophone array and data acquisition system provide accurate records responding to the seismic testing, limitations remain [14]. The widespread application of cable-based acquisition system that has been stymied by the high cost of sensors installation and connection has warranted the emergence of the cost-saving wireless sensing system. Deployment and convenience are the main factors considered for seismic research and application. A wireless sensor is a response to limitations such as a large number of sensing channels and inconvenient deployment [15,16]. For example, high-density land acquisition systems require as many as 50,000 live channels and a receiver density of 2000–3000 channels/km^2^ [17]. For single-sensors, acquisition systems up to 1000–2000 receivers per line will be required with sensor spacing of 5–30 m. These parameters suggest adoption of an acquisition system that does not rely on cables. Furthermore, communication and detection reliability bring the field closer to studies on autonomous sensors that could collect, analyze, or share data “smartly” [18].

If the current trend of evolution continues with wireless sensors, it is likely that seismic monitoring systems will employ new transmission technologies to meet needs at every development stage. Key features of the main solutions for wireless communication are summarized in Table 1. Current wireless communications are grouped as short- and long-range communications. For short-range communication, the theoretical information transmitting range is from 10 to 160 m while the reliable range is less than the expected range considering the harsh environment and weather. For long-range wireless communication, cellular communication no matter 3G or 4G LTE protocol highly relies on the coverage and the construction of cellular network. Moreover, for the three-dimensional (3D) outdoor survey for complex geography, GPS-embedded wireless geophones can position themselves, which benefits data mining and processing.

Thus, this paper proposed a novel satellite-based communication wireless geophone sensor for surface wave testing of soil, which is easy to deploy, cost effective, and expendable. The hardware is designed to be conveniently assembled from a standard high-precision accelerometer and low-cost computer that are easy to get. The software is developed and optimized to fit the satellite communication for the surface wave testing. Thanks to the computational core of the computer, tailored coding algorithms are embedded in both the sensing node and server to efficiently prevent data loss.

In this paper, hardware assembly and connection are first introduced, followed by software development. A compressive sensing algorithm is presented to encrypt the measurements and prevent information loss. An in-site experiment is designed to validate the functionality of the proposed satellite-based wireless sensing node. The approach will also be applied to future studies on large-area surface wave monitoring strategy.

## 2. Hardware Architecture

The hardware of the portable wireless sensing packaging could be separated into four primary parts: A computational computer, a portable battery for power supply, an accelerometer for monitoring ground motion, and a satellite-based communication device.

### 2.1. Computational Core

Raspberry Pi 3 Model B+, a commercial portable single-board computer developed by Raspberry Pi Foundation in the United Kingdom was employed as the computational computer. Its computational core is Broadcom BCM2837B0 with 1.4 GHz 64-bit quad-core processor. The Raspberry Pi 3 Model B+ uses 1 GB synchronous dynamic random access memory (SDRAM) as memory, and the external SD slot makes larger storage possible. It has four USB 2.0 ports, 2.4 GHz and 5 GHz IEEE 802.11.b/g/n/ac wireless LAN, Bluetooth 4.2, 300 Mbps Gigabit Ethernet, and extended 40-pin GPIO header for multiple purpose communication. The operation system (OS) of Raspberry Pi is a Debian-based OS named Raspbian, which is highly optimized for the its tailored central processing unit (CPU). The Raspberry Pi is an ideal general-purpose computer for a wireless sensing node because of the low price (only 35 USD) and aforementioned high performance.

### 2.2. Battery

A USB portable charger power bank made by PISEN with a capacity of 20,000 mAh was employed to provide 5 V up to 2.0 A power supply for the computer and other electric components (Figure 1a). The size of the charger was 152.4 mm × 78.7 mm × 25.4 mm, and the weight was about 1.5 pounds. With six high-density Li–Polymer batteries, the power bank is a good device, retaining up to 70–80% of its full capacity even after 500 charge–discharge cycles. While considering the shipping issues, the sealed lead-acid rechargeable battery is also a good alternative for readers.

### 2.3. Accelerometer

A MEMS (micro-electro-mechanical systems) sensor was chosen to measure the ground motion in the node. The advantages of MEMS sensors include small size, low cost, low power consumption, high accuracy, and high resistance to vibration and shock [19]. The accelerometer was an Adafruit MMA8451Q 3-axis accelerometer sensor. Its voltage range is 3.3–5.0 V with an operating temperature of −40–85 °C. It is a general-purpose sensor for various applications such as navigation, robotics, fitness and well-being, augmented reality (AR), context awareness, smart devices and tablets, and ultra-books. The sensor is quite small with a size of 17.8 mm × 20.3 mm. Its key sensor is the MMA8451Q (made by NXP Semiconductors), blending a low-power accelerometer to offer three degrees-of-freedom sensing for three-dimensional (3D) orientation. The accelerometer provides programmable acceleration ranges of ±2/±4/±8 g with 14-bit data and up to 800 Hz sampling rate, which is highly accurate for sensing the ground motion. This sensor has a digital interface namely I^2^C for communication. Thus, the I^2^C interface was chosen to communicate with the computational computer, as shown in Figure 1a.

### 2.4. Satellite Device

As mentioned previously, wireless communication was achieved using a satellite device, RockBLOCK 9602, with dimensions 76 mm × 52 mm × 19 mm, weighing only 76 grams. The heart of the RockBLOCK 9602 is an Iridium 9602 modem, a commercial two-way satellite data transceiver, with a size of 41.0 mm × 45.1 mm × 13.1 mm. The Iridium 9602 transceiver does not require a SIM card. A GPS receiver is also embedded to provide relatively accurate positioning data. The device interface consists of a serial interface, power input, network available output, and power on/off control line. The Iridium 9602 is able to deliver up to 340 bytes mobile-originated messages and up to 270 bytes mobile-terminal messages via a simple AT command interface. Meanwhile, the two-way UART interface of the satellite device defines the possibilities for the integration of single-board computer, sensor, and server for delivering acquired data and remote controlling.

The RockBLOCK 9602 hosts a small form factor Molex connector to link the ground, power, and signal lines to the USB port of Raspberry Pi 3 Model B+ as shown in Figure 1b. The baud rate of the RockBLOCK 9602 is 19,200. It exposes the modem’s serial interface via a breakout connector over serial and, also, offers an SMA connector for external antenna usage. The antenna frequency range is 1616–1626.5 MHz combining the global coverage of the Iridium satellite constellation (66 satellites in orbit around the Earth) with the low latency of the short burst data (SBD) service to provide highly reliable satellite communications from any point on Earth including the polar regions, marine, and land areas. Its environmental operation temperature range and power consumption is −40–85 °C and 1 Watt, respectively.

### 2.5. Packaging and Connection

To emphasize its ability, a satellite-based wireless sensor was assembled in a waterproof package with all components integrated. The fully assembled wireless node pictured in Figure 1b was connected using a series of wires carrying power and signals. A 200 mm (length) × 150 mm (width) × 100 mm (height) single enclosure with a clear door was employed as the enclosure. The power bank was fastened to a plastic board of the enclosure to directly power the Raspberry Pi computer. The accelerometer sensor was directly connected the I^2^C ports from the GPIO pins of the Raspberry Pi 3 Model B+, and the satellite device was connected to the computer via a USB port using the UART communication interface. Both were two-way communications to ensure the implementation of sending measurements and receiving commands. A highly sensitive accelerometer was mounted on the floor of the enclosure to measure the ground motion. By using an IP66 level plastic outdoor enclosure and six 5 gram silica gel desiccant packages, the whole packaging was waterproofed to ensure the functionality of electric components under severe environments. The entire packaged wireless sensing node is demonstrated in Figure 1c.

## 3. Software Architecture

The success of a wireless sensing solution for surface wave testing relies on processing of data transmitted from the satellite device and the implementation of a cloud-based database for the storage. In response to the aforementioned need, the software architecture embedded into the Raspberry Pi 3 Model B+ was designed to push the acquired data to a server as time series. Pre-programmed codes on the computational core commanded the satellite device to send encrypted data acquired by the sensor in the form of SBD via satellite communication. The software architecture utilized RockBLOCK’s commercial cloud-based server, which is ideally suitable for storage, analytics, and management of data acquired from the wireless sensing nodes. As we can see in Figure 2, each user maintained a client identified by an assigned account and identifier key that were used for accessing messages in the commercial database. Data written from each satellite device was tagged with a unique registration that was used to record the data to a corresponding data port. All data were recorded according to the universal time coordinated (UTC) clock and could be analyzed or processed based on user-defined algorithms for further analytics and decision-making. A pre-trigger strategy was employed to record the entire waveforms caused by the impact of surface wave testing. Based on the UCT tags of each encrypted SBD, uploaded data could be synchronized for surface wave analysis. The whole codes for the actions of acquiring measured ground motion from the accelerometer sensor, encrypting data to SBD message, and transmitting SBD to satellite device were embedded onto Raspberry Pi’s CPU programmed by the Python language.

Meanwhile, considering the transmitting limit of SBD (340 bytes for outgoing and 270 bytes for incoming), a data-compressive method was designed to shorten the transmitting lengths of measurements. The compressive sensing (CS) methodology proposed by Donoho [20] and Candès [21] in 2006 was employed in this research for encrypting and decrypting the acceleration data. Here is a brief summary of CS theory:

Any compressible data defined by a sparse vector can be approximately recovered by convex optimization from random linear measurements that are considerably lesser than the original length.
(1)y=Φx=ΦΨα+n=Θα+n,
where y∈ℝM is the linear measurements, x∈ℝN is the compressible data, α∈ℝN is the *K*-sparse (*K*-nonzero elements) vector representing the original signal x, Ψ∈ℝN×N is a specific basis matrix which transfers signal x to sparse vector α in a specific domain. Φ∈ℝm×N is the random linear measurement matrix (K<m≪N), Θ=ΦΨ represents the transfer matrix from α to y. n denotes the measuring noise and error.

It is highly possible to recover the raw data with the random measurement matrix which is in coherence with the original data ensuring the information is distributed and carried by all the new components without bias.

When the measurements number, *m*, is larger than μ·K·log(N/K) (where μ is a constant, μ≈4), this ill-posed problem can be recovered with high-probability by optimizing the convex problem represented in Equation (2):(2)$$\min{\left||\alpha |\right|}_{1}\;\mathrm{subject}\;\mathrm{to}\;{\left||\Theta \alpha -y|\right|}_{2}<\varepsilon ,$$
where ${\left||\alpha |\right|}_{1}=\sum_{i=0}^{N}\left|\alpha_{i}\right|$, ${\left||\cdot |\right|}_{1}$ and ${\left||\cdot |\right|}_{2}$ represents the $l_{1}$-norm and $l_{2}$-norm, respectively, and ε denotes the noise bound.

Comparing to the traditional sampling-and-sending wireless communicating method, CS-based coding algorithm has the ability of transmitting far lesser encryption data than the original measurements and decrypting the information from received data with an optimization processing [22]. It is important to emphasize that the CS strategy is robust which can protect against lost data packages. For wireless communication, losing data packages is a great threat in transmission which causes unnecessary failures or disasters for monitoring and decision-making applications. Thanks to CS coding strategy, each value of the transmitting vector, y, contains the information of all the raw signal, which reduces the risk of “losing all eggs from one basket”.

The CS-based coding algorithm was embedded in the computational core of the Raspberry Pi’s microcontroller by the Python language, and the decrypting algorithm was coded in MATLAB in the server with the corresponding random matrix. A simple example was tested by a programmable function generator (Tektronix Arbitrary Function Generator model AFG3022C) as a demo. The sampling frequency for generating was 1000 Hz, and the sample rate for sensing was 800 Hz. The time-history signal (Figure 3a) had 10 frequency components, which is sparse in the frequency domain (Figure 3b). It was coded by the aforementioned random linear measurement matrix to form encrypted data of length much smaller (40 points) than the previous one. After transmission to the server, the encrypted data was decoded by $l_{1}$-norm optimization algorithm. As we can see in Figure 3c, the recovered signal (red solid curve) had a good agreement with the initial measurements (color blue), which validates the functionality of the proposed data-compressive method. The compressive ratio was 1:16, which means 5440 bytes of acquired data regarding the ground motion could be transmitted in 340 bytes.

## 4. Applications

The functionality of the sensing node was validated in a real-world experiment of surface wave testing of the soil layer in Harbin. Using the GPS-enable, the proposed sensing node was deployed in an outdoor testing ground of Harbin Institute of Technology as shown in Figure 4. A 4 kilogram sledgehammer was employed to trigger the ground motion by impacting on a polyethylene plate placed on the ground to generate the seismic wave. The surface wave was collected by the wireless sensing node with a sampling rate of 800 Hz. As a proof-of-concept test, only five wireless sensing nodes respectively located at 5, 10, 15, 20, and 25 m from the impact position were employed to acquire the seismic waves. In total 600 points were acquired by the accelerometer in 0.75 s. The raw measurements were encrypted based on the proposed random linear measurement matrix to compress the size of the transmitting data. The compressive data was transmitted via six SBDs by satellite communication. Each SBD contained 60 bytes encrypted information to reduce the possibility of losing data although up to 340 bytes could be transmitted in one message. Meanwhile, the raw measurements were saved to the SD memory card as a backup.

## 5. Results

As we can see, the plain text received from the RockBLOCK 9602 was encrypted by a random linear measurement matrix. To decrypt the data transmitted on the server, $l_{1}$-norm optimization algorithm was employed to reveal the seismic wave acquired by the accelerometer. After the decryption, a comparison between the raw time-history signals recorded by the accelerometer (backup data) and the signal reconstructed from the encrypted SBD transmitted by the satellite device is shown in Figure 5. The encrypted data is shown in Figure 5a, and the original acceleration data recorded in the embedded SD memory card and the reconstructed acceleration data extracted from the cloud server are respectively shown in Figure 5b,c. As we can see, the recovered data has a good agreement with the backup data (the reconstructed errors are demonstrated in Figure 5d, which endorses the functionality of both the hardware architecture and the coding program of the wireless sensing node.

All of the data packages were numbered sequentially for sending, although data package loss seems unstoppable for wireless sensing resulting from multiple reasons, such as weather, environmental interaction, and unclear sky. In one case of the outdoor field testing for sensor 5, one of the six encrypted packages was lost (Figure 6a). Thanks to the CS-based encryption strategy, the original acquired accelerometer data can be preserved through the random coding measurement matrix. As demonstrated in Figure 6a, the data received from the server has one packet lost from the satellite communication. After reassembling the remaining packets by their UTC tags, the data was recovered (as show in Figure 6c) by the $l_{1}$-norm optimization algorithm with a limited error (Figure 6d maximum error of 19.1%). The optimization approach yielded good data recovery results when compared with the backup data (Figure 6b) saved in the SD memory card, which proves the CS-encrypted data is robust even with lost data packets.

The acceleration data recovered from all the five sensors are shown in Figure 7. Meanwhile, GPS information was also transmitted from the wireless sensing node to the server via satellite communication, which provides the possibility of the deployment of a wireless geophone array for large-area site surface wave testing. The GPS information of the wireless sensing node was extracted from the satellite communication and stored in a separate output variable in the database. Based on the longitude and latitude information, the location of this proof-of-concept test was visualized in the map as shown in Figure 8.

## 6. Conclusions

In this paper, a wireless sensing node using satellite communication is proposed for surface wave testing of soil layers. Unlike previous work in cabled geophone, which is connected to the specific seismic instrument and limited by wires, the proposed method uses a MEMS accelerator and wireless sensing to upload the data to the server. In doing so, a low-cost portable sensing node can be achieved as a promising alternative to the expensive instrument. It can be densely and rapidly deployed throughout the measured area. Another important contribution of the study is the application of CS algorithm, an effort against data lost in wireless communication caused by environmental interaction. The recovery from lost data is achieved based on the encrypting algorithm and decrypting $l_{1}$ optimization.

The proposed sensing node consists of one computational core, a portable power supply, MEMS accelerometer for sensing ground motion, and a satellite-based communication device. The functionality of the proposed architecture was validated in the surface wave testing outside of the laboratory. Five of the sensors were employed to measure the ground motion, encrypt the measurements, and upload the data to a cloud-based server via satellite communication. The acceleration data were successfully decrypted with limited error in the server even when up to 16% data was lost during communication.

The proposed computational wireless seismic sensing system is a game-changing technique for its low-cost hardware and software architecture, easy deployment, accurate measurement, and reliable communication. Future work will continue the research on both geological research and earthquake engineering applications. In the future, we will also develop this satellite-based geophone to record a seismogram as a seismometer for long-term monitoring of ground movement.

## Figures and Tables

**Figure 1 sensors-19-04364-f001:**
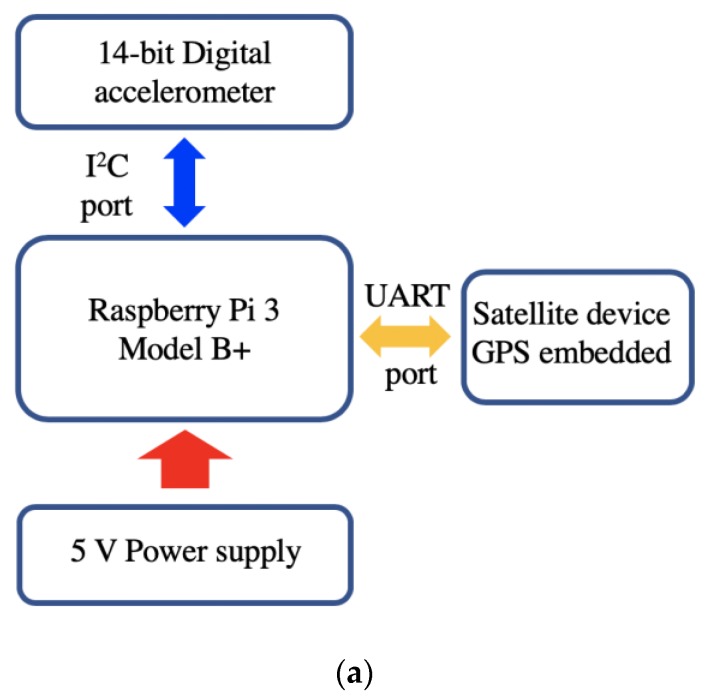
Hardware architecture: (**a**) Hardware design for sensing interface, computational core, wireless communication, and power supply; (**b**) hardware connection with key components highlighted; and (**c**) assembled satellite-based sensing package for in-site surface wave testing.

**Figure 2 sensors-19-04364-f002:**
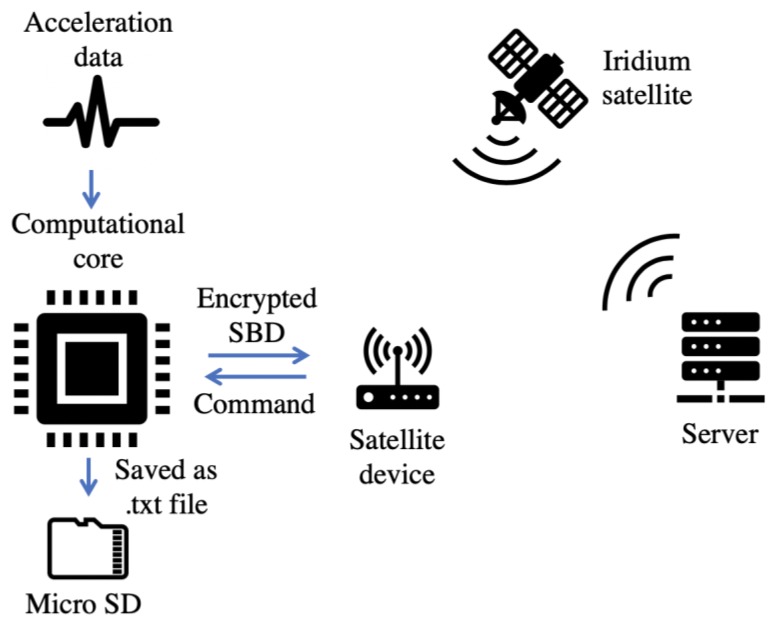
Schematic diagram of software architecture. SBD—short burst data.

**Figure 3 sensors-19-04364-f003:**
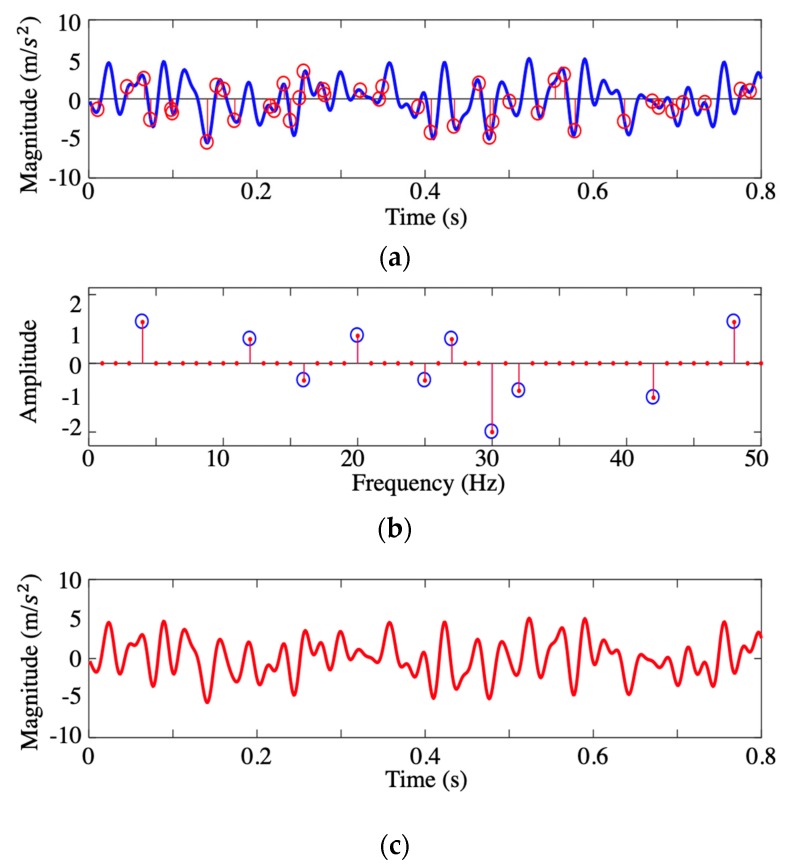
Example of compressive sensing (CS)-based encrypted algorithm: (**a**) Sampling points and original time-history signal containing 10 frequency components; (**b**) reconstructed sparse vector in the frequency domain; and (**c**) recovered signal from the random sampled points.

**Figure 4 sensors-19-04364-f004:**
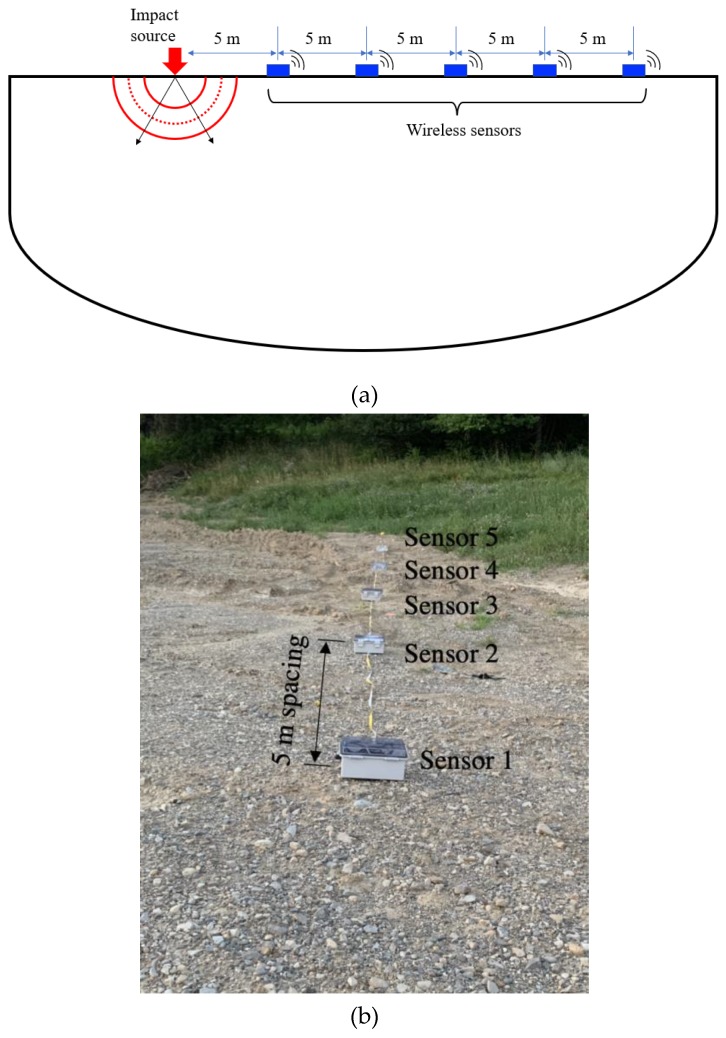
Satellite-based sensing nodes for in-site surface wave testing: (**a**) Schematic of surface wave testing; (**b**) view of testing at the outdoor site.

**Figure 5 sensors-19-04364-f005:**
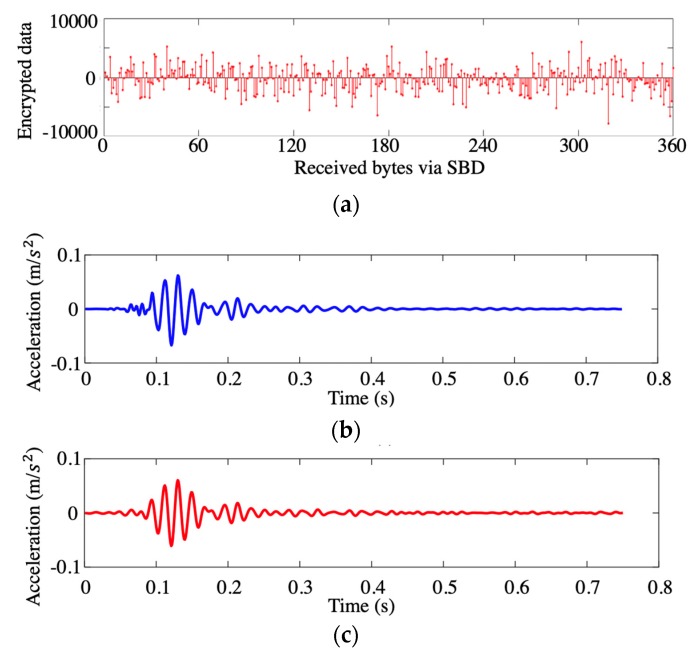
Data recovery results of the measured acceleration by sensor 4: (**a**) Encrypted data; (**b**) original measurements extracted from backup file; (**c**) recovered data from SBD data transferred via satellite communication; and (**d**) comparison and reconstruction error.

**Figure 6 sensors-19-04364-f006:**
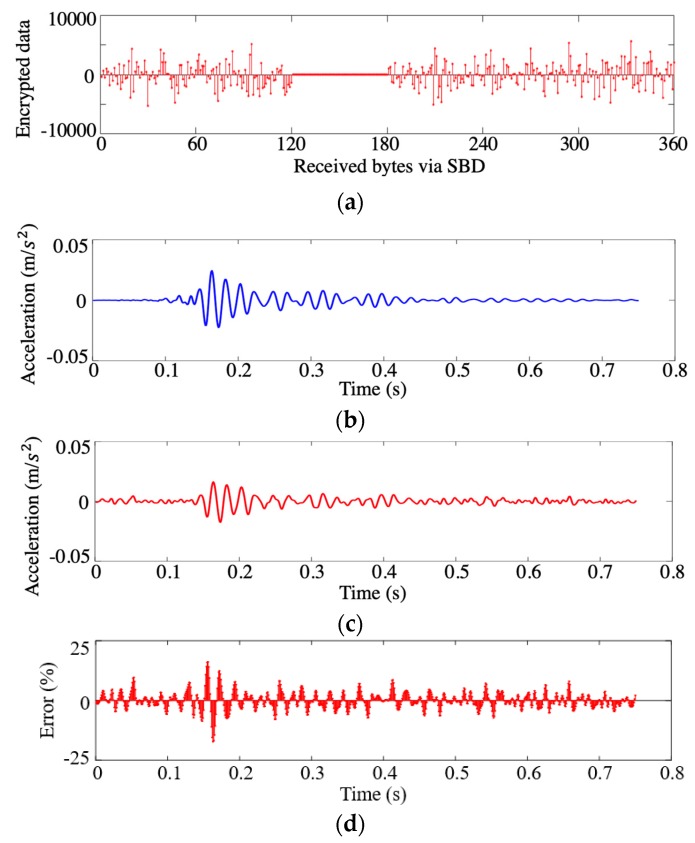
Recovered results of sensor 5 from SBD data loss: (**a**) Received data packets with 16.7% data loss; (**b**) original acceleration extracted from backup file; (**c**) recovered data from the remaining SBD data packets; and (**d**) comparison and reconstruction error.

**Figure 7 sensors-19-04364-f007:**
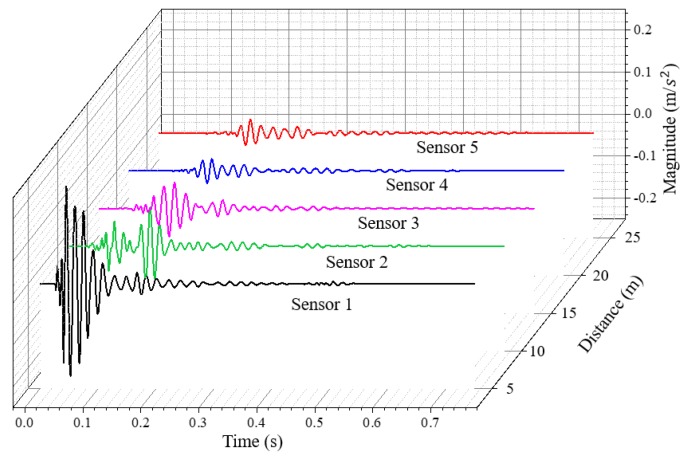
Recovered data of the five satellite-based sensing nodes.

**Figure 8 sensors-19-04364-f008:**
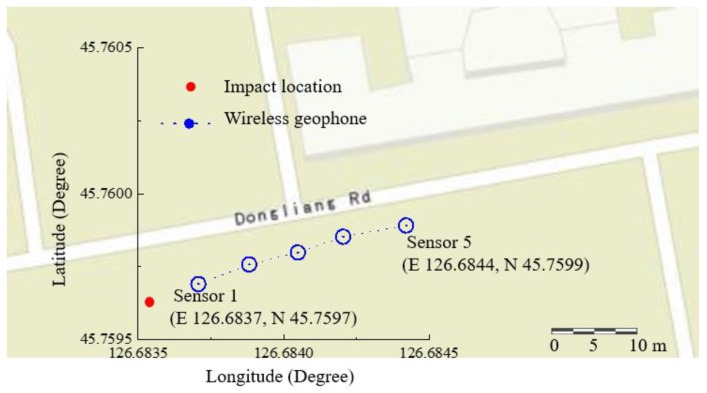
GPS information provided by the satellite-based sensing node.

**Table 1 sensors-19-04364-t001:** Summary of wireless communication.

	Protocol Name	Data Rate	Range	Power Consumption (mJ/Mbyte)	Coverage
Short range	ZigBee	250 kbps	50 m	8−32	-
	Bluetooth	1−2 Mbps	100 m	8	-
	MB-OFDM	53−200 Mbps	10−30 m	8−16	-
	Wi-Fi	6−54 Mbps	70−160 m	120−160	-
Long range	GSM	384 kbps	1−2 km	300,000	Cellular station
	3G	2 Mbps	1−2 km	80,000	Cellular station
	4G LTE	3−8 Mbps	1−2 km	40,000	Cellular station
	Satellite	100−500 kbps	Up to 2000 km	8000	Global

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
