# Peer review of "Satellite-Based Wireless Sensor Development and Deployment Studies for Surface Wave Testing"

_sensors, 2019, doi:10.3390/s19204364_

Round 1
Reviewer 1 Report
This paper is suitable for publishing and I have no further revision requests. I like the use of compressive sensing and the overall design in reducing payload. Similar data compression approaches could potentially be used for distributed acoustic sensing (often burdened by huge data load).
Author Response
Thank you so much for your efforts!
We really appreciate your help!
Reviewer 2 Report
Review of Satellite-based Wireless Sensor Development and Deployment Studies for Surface Wave Testing
The authors test a prototype setup for local site characterization using a small geophone package. They also show an application of this for five sensors. While the paper is mostly well written in terms of explanation, the author’s do need to have this work checked for grammar and language usage. I can appreciate the difficulty of writing in a second language and the author’s have done a relatively good job, but I think it needs more work. Overall the tests and methods seem reasonable. While the scientific impact is likely not huge, it is a valuable contribution and could provide a starting point for others. Therefore, I recommend this work for publication after the grammar and language is dealt with.
Below are a few more comments:
Line 12: you want systems
Line 17: remove the a
Line 25: I don’t follow this sentence.
Line 35: You wand: “have attracted”
Line 107 are these US dollars?
Line 108: Can you explain if this is Lithium? If so then there are large shipping issues so one would need to be careful.
Line 222: Does this compression method have limitations when you get very large signals?
Figure 5: You probably want acceleration not magnitude.
Figure 5 (d): It is hard to see the error since you have such large scales on the vertical axis.
Figure 7: You want acceleration again.
Author Response
Thank you so much for your review, we have revised the manuscript according to your comments. Please find the revised version in the attachment.

Round 2
Reviewer 2 Report
I appreciate the work, done by the authors, to improve their manuscript. I have no concern about the scientific content of the manuscript. I would recommend this for publication assuming they can improve the grammar and language usage. I apologize, but I am unable to correct all of the language and I understand that this is not ideal. Outside of the language usage (which is only grammatical in nature), I recommend this for publication.
There are still a number of grammatical issues. I have in no way pointed out all of these.
E.g.:
line 54: "One common device used for a seismic survey is a geophone that converts ground movement (velocity or acceleration) into a time-history signal in units of voltage."
line 56: "A geophone has played an important role in seismology measurements."